# Host Identity and Consumption Behavior: Evidence from Rural–Urban Migrants in China

**Nianzhai Ma** [1,*] , **Weizeng Sun** [1,*] **and Zhen Wang** [2]

1 School of Economics, Central University of Finance and Economics, Beijing 102206, China
2 School of Statistics, Huaqiao University, Xiamen 361021, China
* Correspondence: 2020110096@email.cufe.edu.cn (N.M.); sunweizeng@gmail.com (W.S.)

**Abstract:** Rural–urban migrants significantly contribute to developing economy, whereas they face high housing prices, rare work opportunities and insufficient consumption. By stimulating the consumption of migrants, their happiness and life satisfaction can increase, regional consumption structural transformation can be stimulated, and economic growth can be boosted. By exploiting the data from the "China Migrants' Dynamic Survey" (CMDS) conducted by the National Health and Family Planning Commission of China, this study explores the effect of rural–urban migrants' host identity on their consumption. We measure host city identity by migrants' sense of belonging in the city. Propensity score matching (PSM), instrumental variable methods (IV), and structural equation modeling (SEM) are adopted to tackle down the potential selection bias and endogeneity concerns. As indicated by the empirical results, host identity significantly impacts rural–urban migrants' consumption, while regional cultural differences hinder migrants from forming host identity. Compared with those without a host identity, migrants with a host identity, the monthly household consumption increased by 4%, and savings decreased significantly by 1.7%. As revealed by the heterogeneity analysis, the host identity effects are significantly larger for migrants aged over 30 years or for those staying in big cities. The results of SEM show that a one-unit increase in the latent variable of identity will increase the consumption by 5.2%, and education, social insurance, and household registration have a significant effect not only on consumption but also on host identity. The findings of this paper contribute to a comprehensive understanding of the psychological and economic integration of migrants in cities and provide valuable suggestions for city managers and policymakers.

**Keywords:** host identity; social integration; consumption behavior; household saving; China Migrants' Dynamic Survey

## 1. Introduction

As industrialization and urbanization are leaping forward, immigration activities have become progressively frequent. According to UN DESA (United Nations, Department of Economic and Social Affairs, United States of America), the number of international immigrants has increased on a year-to-year basis, from approximately 174 million in 2000 to 272 million in 2019, an increase of 56.3% in two decades (Data source: https://www.un.org.development/desa/en, accessed on 17 September 2019). Moreover, the number of rural–urban migrants in developing nations is increasing year by year. As far as China is concerned, rural–urban migrants surged from 121 million in 2000 to 241 million in 2017 (National Health Commission of the RPC: Report on China Migrant Population Development 2018). Rural–urban migrants have enormously contributed to China's economic growth. For instance, they provide the necessary labor supply and facilitate industrialization and industrial agglomeration [1–6]. Besides, their consumption and tax payments in cities turn out to be critical to the urban economy [7,8]. However, they still encounter numerous problems (e.g., high housing prices, low employment and insufficient

consumption), which impact their sense of belonging and life satisfaction, which is of critical significance to their consumption and social integration [9,10].

Identity refers to the sense of who I am, determines who this person is and which group they pertain to [11]. Tajfel [12] proposed that the groups people belong to are a vital source of self-esteem and pride. Identity is complex and can be used in a wide variety of scenarios. Categorization is a crucial step for identity, and identity will be formed via the process of self-categorization. As noted by Jenkins [13], identity is a process of creation. Accordingly, different identity means different forming processes, which will lead to differences in people's behavior and economic performance [14,15].

This study focuses on the host city identity of rural–urban migrants. As is widely known, migrants often face many problems in the host city or country (e.g., language barriers and cultural differences). Besides, migrants are likely to experience discrimination [16]. The mentioned barriers impact the integration of migrants and ultimately impact their identity perception [17]. Identity and perception can impact decision-making, leading to differences in individual behavior and economic outcomes [18–20]. Several researches have indicated that migrants' economic behaviors, such as earnings, probability of being employed, consumption, etc., will be dependent of identity. However, existing research on evaluating the correlation between host identity and consumption mainly focuses on international immigrants and the conclusions are also mixed. It is still doubtful whether rural–urban migrants' consumption, which is a critical factor of the urban economy, is dependent of host identity.

Using data from the "China Migrants' Dynamic Survey" (CMDS) conducted by the National Health and Family Planning Commission, this study studies the effect of rural–urban migrants' host city identity on households' consumption. Host city identity is measured as the feeling of belonging to a particular city. Different variables of host identity are exploited to verify the robustness of the identity effect. As indicated by the empirical results, host identity has significant positive effects on rural–urban immigrant's consumption, the consumption level of rural–urban immigration with host identity is significantly higher than those without host identity. After solving the self-selection bias and potential endogeneity concerns, the results continuous to be robust. Furthermore, the empirical results reveal that the host identity effect is more evident for migrants living in larger cities (over 5 million) or aged over 30 years. The results of SEM show that a one-unit increase in the identity will increase the consumption by 5.2% In addition, SEM shows the relationship between the variables, education, social insurance, and household registration have a significant effect not only on consumption but also on host identity.

The remainder of this study is organized as follows: In Section 2, we review the related literatures. In Section 3, we describe the data, variables, and empirical strategies. In Section 4, we present the main results and discuss the heterogeneities of the effect of host identity. In Section 5, we conduct an SEM analysis. We conclude our paper in Section 6.

## 2. Literature Review

Akerlof and Kranton [19,21] hold that identity is consistent with behavioral norms, and it impacts individual behavior by influencing individual benefits. Identity is complex and can be shaped by any labels (e.g., race, religion, and gender). When an identity is activated, it will impact behavior. More generally, identity and behavior go hand in hand [22]. Identity influences behavior, and behavior in turn shows identity information. People display identity information through their behavior to gain a sense of belonging in the group [23,24]. Behaviors related to identity expression are everywhere. For instance, Xie and Chen's (2022) research shows that Chinese urban residents show their status as urban dwellers by purchasing expensive housing in a good environment, and their results show that different housing dimensions are significantly related to identity expression but in different ways. Specifically, housing conditions are primarily associated with urban identity, whereas housing consistency and housing context are significantly related to local identity [25]. For instance, Kirchmaier et al. [26] found that people with religious

identities have fewer bribery and tax evasion acts, and they are more enthusiastic about volunteer activities than people without religious beliefs. Bisin et al. [27] reported that Muslim immigrants in the UK with higher religious identity levels have a higher probability of intermarriage and lower wages. Identity often impacts consumption since consumers draw upon products to show their identities. The research conducted by Coşgel and Minkler (2004) [28] indicated that religious identity will stimulate believers to consume more religious goods, and this process can express religious piety and address the free-rider problem. Chattaraman and Lennon [29] revealed that the strength of ethnic identity was a significant factor of cultural consumption. Furthermore, some articles have pointed out that in order to express identity, people are more engaged in material consumption than spiritual consumption [30].

Studying migrant identity is actually studying their integration [31,32]. As the number of international migrants has grown robustly, considerable scholars noticed that rural–urban and international migrants encounter the identical problems in the host economy (e.g., low education level, language barriers and low-skilled). The process of host identity formation has been extensively studied. Regional differences in hometown and city hinder the formation of a host identity. For instance, religious, ethnic, and cultural differences hinder migrants from constructing host identity [33–35]. Moreover, the prejudice and hostility of locals are not conducive to host an identity [36–38], but a long-term living willingness and high life satisfaction can help form host identity [36–38].

Identity determines not only the behavior of migrants but also their economic integration [39]. Several studies have indicated that migrants' earnings and probability of being employed will be dependent of identity. For instance, Mason [40] reported that Americans of Mexican and Cuban descent are capable of increasing annual income and hourly wages by acculturating into a non-Hispanic white racial identity. Drydakis [41] revealed that assimilation and integration identities are positively related to immigrant wages, while separation and marginalization identities are negatively related to immigrants' wages after considering various demographic characteristics. Gorinas [42] determined that immigrants who share social norms with the majority experience significantly better employment outcomes, particularly first-generation immigrant women.

Consumption is a key way to show identity, but findings in the empirical field regarding migrants' host identity and consumption remain mixed. Adamopoulou and Kaya [43] treated the European Union enlargement as a natural experiment. They reported that the consumption of immigrant households increased as soon as their home country accessed into the EU. As reported by Dustmann et al. [44], legal identity significantly impacts immigrant consumption; to be specific, documented immigrants consume about 40% more than undocumented. Stodolska and Livengood [45] indicated that immigrants with religious identities have less leisure consumption. However, considerable studies questioned the host identity effect. For instance, Islam and Raschky [46] identified no strong causal relation between immigrants' host identity and performance in the host country's labor market. In addition, Casey and Dustmann [47] considered that host or home identity does not significantly impact immigrant behaviors.

Impacted by several factors (e.g., constraints of education, skills, and job stability), income and consumption inequality have been widespread in Chinese rural–urban migrants. As a consequence, their quality of life is affected, and migrants are isolated from locals, thereby hindering their integration in cities. The household registration system still is an institutional barrier that affects the acquisition of host legal identity by migrants. Even if the Hukou system has been reformed, it is still hindering migrants' welfare by narrowing their access to social security and housing funds offered by the employer [48,49]. The household registration system hinders the source of additional income and benefits for migrants reducing their potential consumption. Chen analyzes the path of Hukou on the consumption of migrant households, on the one hand, migrants do not enjoy local urban hukou, which creates economic insecurity through barriers to employment, social benefits, and health insurance, thus encouraging precautionary savings, on the other hand,

it promotes temporary migration, allowing differences in tastes and values with local urban residents to persist and providing incentives for migrant households to save their transitory income [50]. Chen et al. [51] presented an insight answer for China's low consumption-to-GDP ratio, and their empirical results show that the consumption of migrants without an urban Hukou is 30.7% lower than that of urban locals. As reported by Han et al. [52], Hukou decreases migrants' consumption of livestock products and vegetables and fruit by 8.8% and 4.8%. Wen et al. [53] indicated that the Hukou system causes urban segregation and leads to consumption inequality between rural–urban migrants and urban residents. In addition, the access to public service or long-term stay intention also impacts rural–urban migrants' consumption [54,55].

According to the identity theory, people tend to share the same behaviors, beliefs, and preferences of social identity as the target group and show differences from the off-target group, maintaining consistency with the target group and thus gaining a sense of belonging [56,57]. Akerlof and Kranton extended the scope of economic research by using utility functions to articulate the interrelationship between human identity and behavioral norms [19]. As described earlier, the study of migrants' identity is the study of their integration. In China, with the expansion of urban areas, more and more farmers are moving to cities [58]. Social integration of floating population is a hot issue of common concern for policymakers, city managers, and scholars. Kinds of literature have given detailed studies on the social integration of migrant populations. Consumption as an indicator of the economic integration and utility of the migrant population has been widely studied [59,60]. However, current research on Chinese migrant consumption has focused on household registration, human capital, and social networks, and is rarely focused on the identity effects. In fact, there are considerable kinds of empirical literature focusing on the identity effects of international migrants' consumption, but no consistent conclusions are given [43,47]. This study adds to the literature line by examining the effect of host city identity on Chinese rural–urban migrants' consumption. On the one hand, it helps to understand the urban integration process of China's migrant population. On the other hand, it can provide useful advice to city managers and marketers. More importantly, it helps to find solutions to the problem of low consumption among Chinese residents.

### 3. Data, Variable, and Empirical Strategy

*3.1. Data and Variable Selection*

The data for this study originates from the Social Integration and Psychological Health Individual Questionnaire, a part of the 2014 wave of China Migrants Dynamics Survey (CMDS). CMDS has been conducted annually by the National Health and Family Planning Commission (NHFPC) in China since 2009. Such a survey places stress on migrants aged between 15 and 59 years who have moved to the city for over one month but without local urban Hukou. In each year, a stratified random sample of 100 to 200 thousand migrants are interviewed. Stratified and multi-stage PPS (Probability Proportionate to Size) sampling is adopted as the sampling method, from province to city, towns/districts, as well as communities. Migrants were asked to answer about demographic characteristics, employment conditions, household earning and consumption, access to public health services, as well as medical services.

Social Integration and Psychological Health Individual Questionnaire comply with the 2014 wave of China Migrants Dynamics Survey (CMDS), which was performed in a sub-sample of the big survey (e.g., eight respective cities). The mentioned cities consist of Chaoyang District in Beijing, Jiaxing in Zhejiang Province, Xiamen in Fujian Province, Qingdao in Shandong Province, Zhengzhou in Henan Province, Shenzhen, and Zhongshan in Guangdong Province, as well as Chengdu in Sichuan Province. The sample size of migrants in the respective typical city (district) is 2000, and the total number of data samples is 16,000. In this sub-survey, respondents were asked the questionnaires listed in the primary survey. Furthermore, they were also asked about the feeling of belonging to the city where they now live and other information (e.g., neighborhood composition, access

to public service, as well as local activities). This is convenient for us to measure migrants' host city identity and exam the correlation between host identity and economic behavior.

Migrants' host city identity is measured based on their answer to the questions: (1) Host identity: "Do you agree that you have been a local?" If the migrant answered "Yes" the host identity equals 1, answered "No" equals 0. To verify the robustness, a reverse host identity and two broader host identities are added. (2) Reverse host identity: "Do you agree that you still are a member of your hometown?" If the answer was "Yes", the reverse host identity equals to 1, if "No", it is equated with 0. There were also two broader host identities: (3) Part of the city identity and (4) A member of the city identity: "Do you agree that you are a member or a part of this city?" If the answer was "Totally agree" and "agree", the broader host identity equals 1, if "Disagree" and "Totally disagree", it equals 0.

### 3.2. Empirical Strategy

Multiple linear regression is used to estimate the correlation between host identity and consumption. The baseline specification is written as:

$$y_i = \alpha + \beta \cdot identity_i + \chi \cdot control_i + \lambda_j + \varepsilon_i \tag{1}$$

The dummy variable *identity* is 1 if the immigrants consider they are local and 0 otherwise. Where $y$ is the dependent variable, denoting migrants' average monthly household consumption or savings rate. $\lambda_j$ represents city fixed effects to control for time-invariant differences between cities. $\varepsilon_i$ expresses the error term. $control_i$ represents control variables, which covers a set of individual characteristics (e.g., gender, age, health, education, married statue, Hukou statue, and migrants' participation in social security and medical security programs). Furthermore, household characteristics are concerned (e.g., monthly household income, size of the household population, as well as type of neighbors). Employment features are also controlled, including job occupation, industry, and whether migrant gets free accommodation and meals from work.

### 3.3. Endogeneity and Instrumental-Variable Approaches

There are potential endogenous due to reverse causality and omitted variables though adequate variables are controlled. For example, relevant studies on sociology and marketing hold that identity impacts consumption, while consumption generates identity [61,62]. Besides this, host identity and consumption are dependent of numerous variables (e.g., household assets and credit) [63–65]. The mentioned variables cannot be regulated since relevant data are rare, thereby causing omitted variable bias. For instance, household assets positively impact immigrants' host identity and consumption. If the variable of household assets is omitted, an upward bias will be caused.

A valid strategy to address endogeneity concerns is to use instrumental variables methods. A suitable instrumental variable should be highly correlated with host identity, whereas it does not directly impact consumption. It is generally known that people are inclined to form an identity with people consistent with themselves. Identities are dependent of numerous factors (e.g., language, culture, and religion). On the whole, there are 17 dialects and 105 sub-dialects in different regions of China. Typically, the dialect symbolizes regional culture in China [66]. Based on the mentioned analyses, the dialect distance matrix of the migration of the rural–urban migrants is constructed as an instrumental variable. Dialect distance can be used to measure the dialects differences between the migrants' hometowns and the host cities. The dialectal distance is determined by:

$$Dial\_distance_{pj} = \sum_{m=1}^{M} \sum_{n=1}^{N} s_m^p \cdot s_n^j \cdot d_{mn} \tag{2}$$

where subscripts $m$ and $n$ denote any county in the city $j$ and $i$, respectively. $s_m^p$ represents the proportion of county $m$'s population in city $p$; $s_n^j$ expresses the proportion of county $n$'s population in city $j$. $d_{mn}$ denotes the dialectal distance between county $m$ and county

*n*. Liu et al. [67] identified the dialects for each county of China. By complying with the "Dictionary of Chinese Dialects" and the "Atlas of Chinese Languages", Chinese dialects can fall into three levels, i.e., pan-dialect area, dialect area, and dialect slice. Based on this classification, the dialect distance can be assigned between any two counties: $d_{mn} = 0$ if the two counties pertain to the identical dialect slice; $d_{mn} = 1$ if the two counties belong to the identical dialect area but not the same dialect slice; $d_{mn} = 2$ if the two counties originate from the same pan-dialect area but not the identical dialect area; $d_{mn} = 3$ if the two counties do not pertain to the same pan-dialect area. Accordingly, the cultural distance refers to the weighted average of the dialectal distance between all county pairs across the two cities (the weight represents the population proportion). Its value ranges from 0 to 3, and the larger the variable, the farther the cultural distance will be between the two cities.

In addition, an alternative instrumental variable, migration distance, is adopted. $Distance_{ij}$ indicates the distance between migrant $i$'s hometown and the host city. Migration distance is capable of capturing regional cultural differences and to determine whether to become Tongxiang with the city locals. As a matter of fact, there is a strong culture of Tongxiang (people from the identical village, town, city, or province) in China. Such a type of culture impacts trust and communication [68]. In other words, distance is negatively correlated with the probability of becoming Tongxiang with locals, so it is negatively related to the formation of local identity. The two-stage least squares (2SLS) model is formulated as:

$$\text{First-stage}: \ \widehat{identity}_i = \delta + \eta \cdot IV_i + \varphi \cdot control_i + \lambda_j + \xi_i \tag{3}$$

$$\text{Second-stage}: \ y_i = \alpha + \beta \cdot \widehat{identity}_i + \chi \cdot control_i + \lambda_j + \varepsilon_i \tag{4}$$

where $y_i$ denotes the consumption or savings of migrants, $control_i$ represent all control variables applied in the baseline model, $identity_i$ expresses the endogenous variable of host identity, $IV_i$ is the instrumental variable (e.g., dialect distance and distance), $\lambda_j$ denotes the city fixed effect, $\varepsilon_i$ and $\xi_i$ are the error terms.

## 4. Empirical Results

### 4.1. Descriptive Analysis

Table 1 lists the descriptive statistics of all variables. According to the table, the average monthly household consumption is nearly 3085 yuan, and the average monthly household savings rate reaches 0.486. The monthly average household income is approximately 6432 yuan. As indicated by Figure 1, for the host identity, 3057 rural–urban migrants consider they are locals, in which men take up 13.01%, while women account for 9.41%. Moreover, 10,579 migrants do not think so, in which men account for 45.12%, and women account for 32.46%.

**Table 1.** Descriptive statistics.

| Variable | Definition | N | Mean | Std. Dev. | Min | Max |
|---|---|---|---|---|---|---|
| *Consumption* | Average monthly household expenditure (yuan RMB). | 13,636 | 3085.075 | 3067.3 | 20 | 200,000 |
| *Saving Rate* | Average monthly household savings rate. | 13,636 | 0.486 | 0.193 | −0.5 | 0.992 |
| *Identity* | Whether the migrants think they are city locals. Yes = 1, No = 0. | 13,636 | 0.224 | 0.417 | 0 | 1 |
| *Rev_identity* | Whether the migrants think they are member of hometown. Yes = 1, No = 0. | 13,636 | 0.874 | 0.332 | 0 | 1 |
| *Part_identity* | Whether the migrants think they are a part of the city. Yes = 1, No = 0. | 13,636 | 0.909 | 0.287 | 0 | 1 |

**Table 1.** *Cont.*

| Variable | Definition | N | Mean | Std. Dev. | Min | Max |
|---|---|---|---|---|---|---|
| *Memb_identity* | Whether the migrants think they are a member of the city. Yes = 1, No = 0. | 13,636 | 0.882 | 0.323 | 0 | 1 |
| *Income* | Average monthly household earning (yuan RMB). | 13,636 | 6432.261 | 6944.518 | 800 | 300,000 |
| *Family_size (FZ)* | Total family population. | 13,636 | 2.825 | 1.232 | 1 | 9 |
| *Age* | Age(years) | 13,636 | 32.871 | 8.656 | 15 | 60 |
| *Hukou* | Registered residence, Agricultural Hukou = 0, Non-agricultural Hukou = 1. | 13,636 | 0.138 | 0.345 | 0 | 1 |
| *Bachibaozhu (BZ)* | Whether the migrants get free shelter and food from work, Yes = 1, No = 0. | 13,636 | 0.198 | 0.398 | 0 | 1 |
| *Education* | Educational attainment, 4 dummy variables: 1. with no school education and elementary school 2. middle school and high school 3. college degree in specialty and general college degree 4. Postgraduate. | 13,636 | 1. 9.3% 2. 76% 3. 14.2% 4. 0.5% | | | |
| *Ethnic* | | 13,636 | 0.965 | 0.183 | 0 | 1 |
| *Male* | Male = 1, Female = 0. | 13,636 | 0.581 | 0.493 | 0 | 1 |
| *Marriage* | Married = 1, Other situations = 0. | | | | | |
| *Medicinsur (MI)* | | 13,636 | 0.89 | 0.312 | 0 | 1 |
| *Socialsecur (SS)* | Whether the floating population have any social insurance, Yes = 1, No = 0. | 13,636 | 0.764 | 0.425 | 0 | 1 |
| *Health* | Health status, 5 dummy variables:1. Very good 2. Good 3. Well 4. Not bad 5. Very bad | 13,636 | 1. 26.8% 2. 35.3% 3. 27.0% 4. 10.6% 5. 3% | | | |
| *Neighborhood type (NT)* | Composition of neighbors, 4 dummy variables: 1. mostly migrants 2. mostly local residents 3. balanced shares of migrants and local residents 4. not sure. | 13,636 | 1. 42.7% 2. 21.5% 3. 29.8% 4. 6.1% | | | |
| *Vocation* | Denoting the major occupation, 18 dummy variables. | 13,636 | | | | |
| *Work_unit* | Denoting the type of work unit, 20 dummy variables. | 13,636 | | | | |
| *City* | Cities, 8 dummy variables: 1. Zhongshan 2. Xiamen 3. Jiaxing 4. Beijing 5. Chengdu 6. Shenzhen 7. Zhengzhou 8. Qingdao. | 13,636 | | | | |
| *Dialect distance* | Expressed from 0 to 3. | 13,631 | 1.9551 | 0.91026 | 0 | 3 |
| *Distance* | Distance from hometown to city (KM) | 13,624 | 558.422 | 475.391 | 0 | 3515.2 |

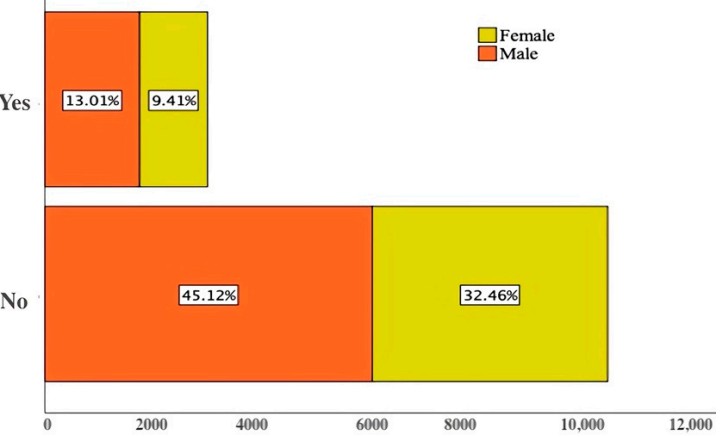

**Figure 1.** Do you agree that you have been a local?

Table 2 lists the migrants' household consumption and savings rate sorted by host identity, which indicates that migrants with host identity are likely to consume more every month in the city compared with those migrants without host identity. However, the mentioned differences do not directly indicate a causal relationship between host identity and the household consumption of the migrants, and more efforts are required to test the causal relationship.

**Table 2.** Two-sample *t*-test.

| | Mean | | Difference | *t*-Test |
|---|---|---|---|---|
| | **Host Identity (1)** | **Without Host Identity (0)** | **Mean (1)-Mean (0)** | |
| *Consumption (RMB)* | 3518.460 | 3033.478 | 484.982 *** (0.000) | 8.391 |
| *Saving rate* | 0.460 | 0.482 | −0.022 *** (0.000) | −5.810 |

Note: Robust standard errors are reported in parentheses. *** $p < 1\%$.

### 4.2. Main Results

This sub-section examines the effect of identity on consumption and savings rate. Table 3 lists the results estimated by using multiple linear regression, the log value of consumption as the dependent variable. We added control variables step by step. In Table 3, Column (1) presents the most parsimonious effects specification controlling nothing, Column (2) adds individual characteristics, and Column (3) illustrates the whole specification, including individual characteristics and city fixed effect. Column (1) shows that host city identity significantly increases the consumption of migrants. Given the estimates, a host city identity causes a 14.3% increase in the consumption of rural–urban migrants. Adding the control variables sequentially in Column (2) and Column (3) does not significantly change the magnitude and significance of identity effects. Overall, having a sense of host city identity will increase the consumption of migrants by 4.4% compared to those who without a sense of host identity. On average, the host identity effect will stimulate the monthly consumption of rural–urban migrants by 135.7 yuan (RMB). In addition, there is a significant positive effect of income, household size, and Hukou on consumption, while the effect of age on consumption is significantly negative.

### 4.3. Robustness Checks

#### 4.3.1. Alternative Measures of Host Identity

Next, we replaced the proxy variables of host identity for robustness checks. Broader host city identity is exploited to estimate identity effects in Table 4. As indicated by the results, for migrants considering that they are a part of the city, their consumption will increase by nearly 2.7%, significance at the 5% level. For migrants considering that they are a member of the city, their consumption will increase by nearly 1.9%, significance at the 10% level. The above results suggest that having a sense of local belonging will increase the consumption of the migrants in the city. Does it imply that not having a sense of local belonging will reduce consumption? We conducted further tests to verify the identity effect. According to the third column of Table 4, we focused on the reverse host city identity. Reverse host identity refers to the migrants' sense of belonging to their hometown rather than their current city of residence. As revealed by the results, reverse identity has a significantly negative effect on consumption. Rural–urban migrants with a hometown identity can significantly reduce their monthly consumption by 3.7% in city. The regression results show that identity perception plays an essential role in the consumption of migrants, and the stronger the sense of host city belonging, the more they consume.

**Table 3.** The effects of host identity on migrants' consumption.

| | Dependent Variable: ln(*Cons*) | | |
|---|---|---|---|
| | **(1)** | **(2)** | **(3)** |
| *Identity* | 0.143 *** | 0.056 *** | 0.044 *** |
| | (0.013) | (0.008) | (0.008) |
| ln (*Income*) | | 0.724 *** | 0.654 *** |
| | | (0.011) | (0.019) |
| ln (*FZ*) | | 0.181 *** | 0.147 *** |
| | | (0.020) | (0.015) |
| *Age* | | −0.003 *** | −0.002 *** |
| | | (0.001) | (0.001) |
| *Hukou* | | 0.119 *** | 0.071 *** |
| | | (0.010) | (0.011) |
| *BZ* | | YES | YES |
| *Education* | | YES | YES |
| *Nation* | | YES | YES |
| *Married* | | YES | YES |
| *Male* | | YES | YES |
| *MI* | | YES | YES |
| *SS* | | YES | YES |
| *Health* | | YES | YES |
| *NT* | | YES | YES |
| *Vocation* | | YES | YES |
| *Job* | | YES | YES |
| *City FE* | | | YES |
| *Constant* | 7.825 *** | 1.577 *** | 2.327 *** |
| | (0.013) | (0.109) | (0.175) |
| *N* | 13,636 | 13,636 | 13,636 |
| $R^2$ | 0.0735 | 0.617 | 0.626 |

Note: Robust standard errors are reported in parentheses. *** $p < 1\%$.

**Table 4.** Robustness checks on the effects of host identity on migrants' consumption.

| | Dependent Variable: ln(*Cons*) | | |
|---|---|---|---|
| | **(1)** | **(2)** | **(3)** |
| *Part_identity* | 0.027 ** | | |
| | (0.012) | | |
| *Memb_identity* | | 0.019 * | |
| | | (0.011) | |
| *Rev_identity* | | | −0.037 *** |
| | | | (0.010) |
| ln (*Income*) | 0.655 *** | 0.655 *** | 0.655 *** |
| | (0.019) | (0.019) | (0.019) |
| ln (*FZ*) | 0.149 *** | 0.149 *** | 0.147 *** |
| | (0.015) | (0.015) | (0.015) |
| *Age* | −0.002 *** | −0.002 *** | −0.002 *** |
| | (0.001) | (0.001) | (0.001) |
| *Hukou* | 0.074 *** | 0.074 *** | 0.073 *** |
| | (0.011) | (0.011) | (0.011) |
| *Control* | YES | YES | YES |
| *City FE* | YES | YES | YES |
| *Constant* | 2.310 *** | 2.315 *** | 2.365 *** |
| | (0.176) | (0.175) | (0.175) |
| *N* | 13,636 | 13,636 | 13,636 |
| $R^2$ | 0.626 | 0.626 | 0.626 |

Note: Robust standard errors are reported in parentheses. * $p < 10\%$, ** $p < 5\%$, *** $p < 1\%$.

### 4.3.2. Saving Rates

Generally, an increase in consumption implies that savings need to be shifted to consumption, which will reduce the savings rate [69,70]. Therefore, this paper uses saving rates as the explanatory variable to verify the identity effect in Table 5. The results in the first three columns of Table 5 show a significant negative relationship between city belonging and savings. It means that the stronger the migrants' sense of city belonging, the higher their consumption, eventually decreasing their savings. The fourth column of Table 5 focuses on the effect of reverse host city identity (hometown identity) on savings. The results show that having a hometown identity increases migrants' savings in the city, implying that migrants do not spend too much in the city when they do not have a sense of belonging. Reverse host identity refers to the migrants' sense of belonging to their hometown rather than their current city of residence. When migrants have a strong sense of belonging to their hometown, they may not live in the city for a long time [59,71]. As a result, they do not spend too much on expensive durable goods in the city but save their income to send to their families or to use later.

**Table 5.** The effects of host identity on migrants' savings.

| | *Dependent Variable: Saving Rates* | | | |
|---|---|---|---|---|
| | **(1)** | **(2)** | **(3)** | **(4)** |
| *Identity* | −0.017 *** | | | |
| | (0.004) | | | |
| *Part_identity* | | −0.007 | | |
| | | (0.005) | | |
| *Memb_identity* | | | −0.008 * | |
| | | | (0.004) | |
| *Rev_identity* | | | | 0.018 *** |
| | | | | (0.005) |
| ln (*Income*) | 0.125 *** | 0.125 *** | 0.125 *** | 0.125 *** |
| | (0.004) | (0.004) | (0.004) | (0.004) |
| ln (*FZ*) | −0.057 *** | −0.057 *** | −0.057 *** | −0.057 *** |
| | (0.006) | (0.006) | (0.006) | (0.006) |
| *Age* | 0.001 *** | 0.001 *** | 0.001 *** | 0.001 *** |
| | (0.000) | (0.000) | (0.000) | (0.000) |
| *Hukou* | −0.032 *** | −0.034 *** | −0.034 *** | −0.033 *** |
| | (0.005) | (0.005) | (0.005) | (0.005) |
| *Control* | YES | YES | YES | YES |
| *City FE* | YES | YES | YES | YES |
| *Constant* | −0.635 *** | −0.630 *** | −0.630 *** | −0.654 *** |
| | (0.049) | (0.049) | (0.049) | (0.049) |
| *N* | 13,636 | 13,636 | 13,636 | 13,636 |
| $R^2$ | 0.217 | 0.215 | 0.215 | 0.216 |

Note: Robust standard errors are reported in parentheses. * $p < 10\%$, *** $p < 1\%$.

### 4.3.3. PSM Results

The previous results suggest that an increase in the sense of local belonging boosts the consumption of the migrants. However, some confounding variables impact the host identity and consumption (e.g., income and Hukou type). For instance, migrants with high income or a local Hukou are more likely to consume more and form a host identity, so migrants without host identity are not comparable to those with host identity. Accordingly, we may wrongly consider the differences between the groups as a result of the host identity effects when it is likely to be attributed to differences between the treatment (migrants with host identity) and control (migrants without host identity) groups. To ensure the robustness of the conclusions we use a propensity score matching for the analysis. Propensity score matching (PSM) methods are exploited to address this selection bias issue. The critical point of PSM is to make treatment (migrants with host identity) and control (migrants without host identity) groups more similar. First, a logit regression model is adopted to

estimate the likelihood of migrants to form a host identity, which includes all observed characteristics. Based on the logit regression results, a propensity score can be determined for each migrant. Subsequently, the most similar control group is found for the treatment group by propensity score. According to Figures 2 and 3, there are significant differences between treatment and control groups before matching. After matching, the two groups turn out to be more consistent with each other. Figure 4 illustrates the kernel density distribution of the propensity scores, and it indicates that most observations are in common support.

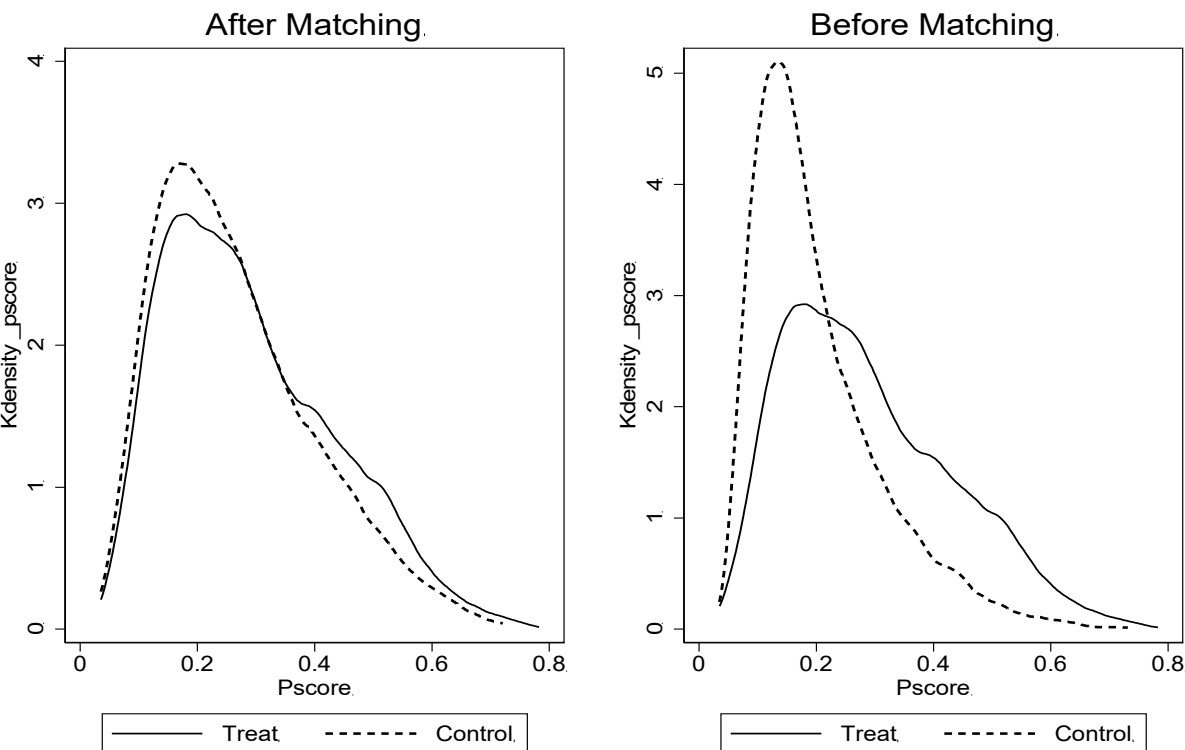

**Figure 2.** Propensity Score Kernel Density.

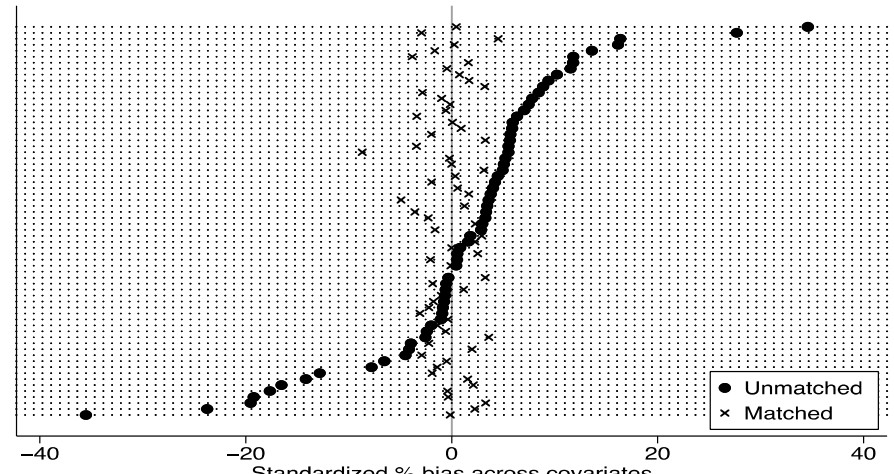

**Figure 3.** Standardized deviation of all control variables.

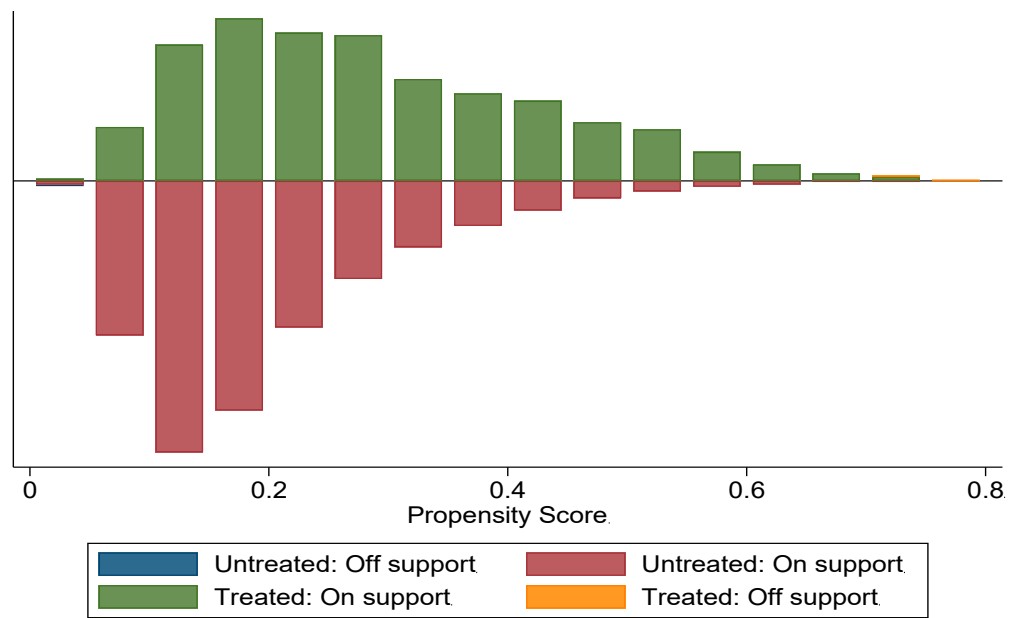

**Figure 4.** Common Support. Note: Figures 2 and 3 based on 1:1 nearest neighbor PSM methods.

After matching, host identity can be treated as a quasi-natural experiment. Table 6 lists the results of ATT (Average Treatment Effect on the Treated). Different rules are applied (e.g., nearest neighbor, kernel, and local linear matching). In the first row, the effect of host identity on migrants' consumption is examined. The results indicate that forming a host identity will lead to migrants' consumption increase by 4%, which is at least significant at the 5% level. According to the second row of Table 4, the effect of host identity on migrants' savings rate is tested. A host city identity will cause migrants' savings rate to decrease by 1.6%, which is significant at the 1% level. It can be found that the ATTs are significantly close to our bassline multiple linear regression results, which truly strengthens the key findings here.

**Table 6.** Average treatment effect on the treated of host city identity.

|  | Matching Rules | | |
|---|---|---|---|
|  | **Nearest Neighbor (1: 4)** **(1)** | **Kernel** **(2)** | **Local Linear** **(3)** |
| ln (*Consumption*) | 0.045 ** (0.030) | 0.041 *** (0.000) | 0.046 *** (0.012) |
| *N* | 13,623 | 13,623 | 13,623 |
| *Savings Rate* | −0.017 *** (0.000) | −0.017 *** (0.000) | −0.016 *** (0.000) |
| *N* | 13,623 | 13,623 | 13,623 |

Note: Robust standard errors are reported in parentheses. ** $p < 5\%$, *** $p < 1\%$.

### 4.4. Instrumental Variable Estimation

Potential selection bias attributed to observable variables has been tackled down. However, there are endogeneity concerns under unobservable variables (e.g., reverse causality and omitted variables). To solve the concern above, two instruments are further employed for host identity. Table 7 shows the two-stage least square estimates of the correlation between the rural–urban migrants' host city identity and two instrumental variables. According to the first column of Table 7, dialect distance positively impacts rural–urban migrants' host city identity, which is significant at the 1% level. In other words, rural–urban migrants are more likely to form a host identity if they have similar

dialects with city residents. The first stage Cragg-Donald Wald F statistic is 60, which demonstrates that this instrument is not weak [38]. Columns (2) and (3) of Table 7 are the second stage that estimates the impacts of host identity on migrants' consumption and savings rate. Compared with those without host identity, migrants with host identity show an increase in their consumption by 39.1% and a decrease in their savings rate by 25.8%. The coefficients are significant at the 1% level. Further, we also find that rural–urban migrants' host identity and consumption are dependent of income, Hukou type, family size, and age. Column (4) of Table 7 presents the estimate of using an alternative instrumental variable. We found migration distance also positively impacts rural–urban migrants' host city identity, which is significant at the 1% level. The results based on migration distance are significantly consistent with dialect distance, which demonstrates that the conclusions of this study are robust. The estimate employing the two instruments is significantly larger (approximately ten times) than the estimate of the bassline model. The possible reason is that a variable negatively related to host identity is omitted. In general, another possible explanation is that the IV estimates exert local average treatment effects of host identity, while multiple linear regression indicates the average treatment effect of host identity over the entire population. In brief, this study reveals that migrants with host identity are likely to consume significantly more than their fellows without host identity.

**Table 7.** The effects of identity on consumption from the 2SLS regression model.

| | First-Stage | Second-Stage | | First-Stage | Second-Stage | |
|---|---|---|---|---|---|---|
| | *Identity* (1) | ln(*Cons*) (2) | *Saving Rate* (3) | *Identity* (4) | ln(*Cons*) (5) | *Saving Rate* (6) |
| *Dial_distance* | −0.046 *** | | | | | |
| | (0.006) | | | | | |
| *Distance* | | | | −0.043 *** | | |
| | | | | (0.008) | | |
| *Identity* | | 0.391 *** | −0.258 *** | | 0.430 ** | −0.289 *** |
| | | (0.136) | (0.064) | | (0.197) | (0.094) |
| ln (*Income*) | 0.026 *** | 0.646 *** | 0.131 *** | 0.025 *** | 0.647 *** | 0.131 *** |
| | (0.008) | (0.008) | (0.004) | (0.008) | (0.009) | (0.004) |
| ln (*Members*) | 0.030 ** | 0.137 *** | −0.049 *** | 0.031 ** | 0.132 *** | −0.047 *** |
| | (0.014) | (0.015) | (0.007) | (0.014) | (0.016) | (0.008) |
| *Age* | 0.002 *** | −0.003 *** | 0.001 *** | 0.002 *** | −0.003 *** | 0.001 *** |
| | (0.001) | (0.001) | (0.000) | (0.001) | (0.001) | (0.000) |
| *Hukou* | 0.082 *** | 0.042 *** | −0.013 * | 0.085 *** | 0.040 * | −0.010 |
| | (0.012) | (0.016) | (0.008) | (0.011) | (0.020) | (0.010) |
| *Control* | YES | YES | YES | YES | YES | YES |
| *City FE* | YES | YES | YES | YES | YES | YES |
| *Constant* | 0.074 | 2.324 *** | −0.632 *** | 0.010 | 2.326 *** | −0.633 *** |
| | (0.107) | (0.106) | (0.050) | (0.107) | (0.103) | (0.048) |
| *N* | 13,631 | 13,631 | 13,631 | 13,624 | 13,624 | 13,624 |
| $R^2$ | 0.089 | 0.581 | 0.597 | 0.0828 | 0.570 | 0.5716 |
| *Cragg-Donald Wald F* | 60.352 | - | - | 29.248 | - | - |

Note: Robust standard errors are reported in parentheses. * $p < 10\%$, ** $p < 5\%$, *** $p < 1\%$.

### 4.5. Heterogeneous Analysis

Due to the differences of rural–urban migrants, host identity may have heterogeneous effects on different situations. In terms of rural–urban migrants, the proportion of inter-provincial migration reached 54.82%, and the proportion of intra-provincial migration was 45.18%. Moreover, half of the migrants live in cities with the population of over 5 million (e.g., Qingdao, Shenzhen, Chengdu, and Beijing). Migrants are divided into two categories

by complying with their characteristics, and then the impacts of host identity on their consumption are tested. Panel A of Table 8 reports the subsample estimates for interprovincial and intra-provincial migration. It can be indicated that the significance of coefficient does not differ between different migration types. Panel B of Table 8 presents subsample estimates for different cities. We find that for those migrants who live in cities with over 5 million people and have a host identity, their consumption will increase by 6.1%, and savings will decrease by 2.7%, which are all significant at the 1% level. However, the host identity effects are significantly smaller for those migrants who live in cities with under five million people. With a host identity, migrants' consumption will increase by 3.6%, which is significant at the 5% level, and savings will decrease by 1.2%, which is just significant at the 10% level. A possible explanation is that the consumption ability of locals in big cities is high. Migrants are likely to spend more consolidating their host city identity in big cities to make themselves look like locals.

**Table 8.** Heterogeneous effects of Identity on consumption by migration type and city.

| Panel A | Inter-Provincial Migration | | Intra-Provincial Migration | |
|---|---|---|---|---|
| | ln (*Cons*) (1) | *Saving Rate* (2) | ln (*Cons*) (3) | *Saving Rate* (4) |
| *Identity* | 0.057 *** | −0.018 *** | 0.043 *** | −0.022 *** |
| | (0.015) | (0.006) | (0.013) | (0.006) |
| *Control and City FE* | YES | YES | YES | YES |
| *Constant* | 2.810 *** | −0.311 ** | 1.238 *** | −0.270 ** |
| | (0.841) | (0.132) | (0.298) | (0.125) |
| *N* | 7364 | 7364 | 6272 | 6272 |
| $R^2$ | 0.576 | 0.260 | 0.688 | 0.208 |
| Panel B | City population < 5 million | | City population > 5 million | |
| *Identity* | 0.036 ** | −0.012 * | 0.061 *** | −0.027 *** |
| | (0.015) | (0.006) | (0.013) | (0.005) |
| *Control and City FE* | YES | YES | YES | YES |
| *Constant* | 2.566 *** | −0.223 | 1.687 *** | −0.434 *** |
| | (0.941) | (0.140) | (0.292) | (0.105) |
| *N* | 6919 | 6919 | 6717 | 6717 |
| $R^2$ | 0.572 | 0.283 | 0.659 | 0.206 |

Note: Robust standard errors are reported in parentheses. * $p < 10\%$, ** $p < 5\%$, *** $p < 1\%$.

Table 9 lists the heterogeneous effects of identity on consumption by individual characteristics. Panel A of Table 9 reports the subsample regression results by gender. It can be indicated that the host identity effects between males and females are very similar. Panel B of Table 9 reports the regression results of subsamples by migrants' schooling. According to the above table, no significant difference is found between different schooling in host identity effects. However, as is shown in Panel C of Table 9, we find the host identity effects vary at different ages. For migrants aged over 30 years, the host identity effect has a greater impact, with a host identity, their consumption increased by 5.7%, and savings decreased by 2.6%, both at the 1% significance level. For migrants under 30, a host identity will stimulate their consumption by 3.2%, which is significant at the 5% level, whereas the effects on saving are not statistically significant. In other words, for migrants aged over 30 years, the host identity effects are significantly greater and significant. The possible reason is that migrants aged over 30 years may stay longer in cities, thereby making them more likely to form a host identity or develop the same consumption habits as the locals.

**Table 9.** Heterogeneous effects of identity on consumption by individual characteristics.

| Panel A | Male | | Female | |
|---|---|---|---|---|
| | ln(*Cons*) (1) | *Saving Rate* (2) | ln(*Cons*) (3) | *Saving Rate* (4) |
| *Identity* | 0.045 *** | −0.019 *** | 0.049 *** | −0.021 *** |
| | (0.013) | (0.006) | (0.015) | (0.006) |
| *Control and City FE* | YES | YES | YES | YES |
| *Constant* | 1.790 *** | −0.429 *** | 2.553 ** | −0.177 |
| | (0.304) | (0.107) | (1.010) | (0.143) |
| *N* | 7927 | 7927 | 5709 | 5709 |
| $R^2$ | 0.643 | 0.216 | 0.609 | 0.271 |
| Panel B | ≥Senior middle school | | <Senior middle school | |
| *Identity* | 0.048 *** | −0.022 *** | 0.048 *** | −0.018 *** |
| | (0.014) | (0.006) | (0.013) | (0.006) |
| *Control and City FE* | YES | YES | YES | YES |
| *Constant* | 2.523 *** | −0.760 *** | 2.113 ** | −0.086 |
| | (0.325) | (0.104) | (0.874) | (0.131) |
| *N* | 5400 | 5400 | 8236 | 8236 |
| $R^2$ | 0.686 | 0.249 | 0.571 | 0.230 |
| Panel C | Age ≥ 30 | | Age < 30 | |
| *Identity* | 0.057 *** | −0.026 *** | 0.032 ** | −0.011 |
| | (0.012) | (0.005) | (0.016) | (0.007) |
| *Control and City FE* | YES | YES | YES | YES |
| *Constant* | 2.356 *** | −0.265 ** | 2.386 *** | −0.720 *** |
| | (0.767) | (0.115) | (0.337) | (0.133) |
| *N* | 8101 | 8101 | 5535 | 5535 |
| $R^2$ | 0.574 | 0.239 | 0.644 | 0.258 |

Note: Robust standard errors are reported in parentheses. ** $p < 5\%$, *** $p < 1\%$.

## 5. Structural Equation Modeling (SEM)

The results of the regression analysis indicate that host city identity has a significant effect on the consumption behavior of the migrants in city. However, these results still have potential limitations. On the one hand, although this work has used two-stage least squares (2SLS) to mitigate measurement error in the definition of host city identity, using observed variables to directly define the host city identity of migrants may still suffer from potential measurement errors. On the other hand, the regression analyses only give the effects of the variables on consumption, but do not show the interrelationships between the variables. The use of structural equation modeling (SEM) can effectively alleviate the problem of measurement error in the definition of latent variables and also present the interrelationship between the variables. It is inconvenient for us to understand the identity transformation (citizenship) and behavior of migrants in the city. Therefore, this paper applies SEM for further analysis.

### 5.1. Structural Equation Modeling (SEM) Theoretical Basis

This work focuses on the impact of host city identity on consumption behavior of China rural–urban migrants. Economics literature states that an individual's income has a significant impact on his or her consumption [72,73]. At the same time, income as an indicator of economic integration of the migrant in host city also has an impact on local identity [74]. In recent years, all major cities in China have been actively competing for talents by offering various facilities for highly skilled migrants to integrate into city. This means that the level of education facilitates the urban integration of Chinese migrants. In further, years of schooling, as a signaling indicator of human capital, have a release effect on

consumption by affecting the quality of employment of migrants (e.g., income, employment stability, etc.) [75]. China has a strict population management system called the Hukou system. The Hukou system legally defines an individual's regional affiliation, and this legal affiliation affects people's access to public services and public resources in a given city [76]. The constraints of the household registration system result in the migrants not being able to easily acquire a legal identity in city [77], which will undermine the migrants' host city identity perception. At the same time, the household registration system is also limiting the access of migrants to public services and thus hindering their consumption [78].

The previous research literature also indicates that the household registration system is an important factor in limiting the consumption of the migrants, and obtaining an urban Hukou identity can significantly increase various consumption of migrants [79,80]. There is no doubt that the availability of social security and health insurance symbolizes the migrants' access to local public resources and has a role in their perception of city identity. More importantly, such social security programs can help reduce the uncertainty of their expenditures, which will eventually affect the amount of other consumption as well. Over the past few years, as large cities continue to introduce various inclusive integration policies, more and more migrant families are migrating to cities. The increase in household size enhances the migrants' sense of belonging in the city [81], while also increasing spending on house rent, food, and entertainment.

### 5.2. Structural Equation Modeling (SEM) and Data Source

Based on the above theories, this paper constructs a structural equation model (SEM) model to explore the effect of host city identity on migrants' consumption and the interrelationship between the variables. With reference to previous studies on consumer behavior, health status, gender, age, and marital status are considered in the model.

Data for the analysis also originate from the Social Integration and Psychological Health Individual Questionnaire, a part of the 2014 wave of China Migrants Dynamics Survey (CMDS). To simplify the analysis, migrants' household income and total monthly consumption is divided into 10 levels, with higher levels representing correspondingly higher income and consumption. Table 10 presents the descriptive statistics of the variables corresponding to the structural equation model (SEM). According to the feeling of host city identity, the analysis shows that migrants have a weaker feeling of local identity, while a stronger sense of membership and participation. The income level of migrants is 5.287 and the expenditure level is 5.269, basically maintaining a balance between income and expenditure.

By analyzing the statistical information, we also found that the average age of the migrants was around 33 years old, and they were in good physical condition (mean health score = 3.76). Within the sample, 13.9% of the migrants have urban household registration Hukou, and the average size of floating families is 2.9. The participation rate of social security and medical insurance programs is relatively higher, of which 75.1% of the migrants participated in social security programs and 87.9% of the migrants participated in medical insurance programs.

According to the above theory and data, this paper establishes the SEM. The conceptual model of this paper of the effect of host city identity on consumer behavior through STATA software is displayed in Figure 5.

### 5.3. Common Method Bias (CMB)

Common method bias (CMB) problem is prevalent in the data collection process of behavioral research. CMB is usually related to the process of data collection, such as the content of specific items, scale type, response format, and the general context. Typically, CMB is a systematic bias, which means that if there is a common method bias will result in a common bias in all the variables collected. Systematic error variance can have a serious confounding influence on empirical results, yielding potentially misleading conclusions.

Procedural control and statistical control are effective ways to mitigate CMB. Regarding procedural control, the researcher needs to control the type of questionnaire, data collection environment, etc., from the beginning of data collection. Researchers usually obtain predictor and criterion variables from different data sources to avoid the problem of common method bias [82]. In addition, statistical tests are often used in the model construction phase to test for potential CMB troubles. Since the data in this paper come from large survey data (CMDS), we cannot control the data collection process at the beginning, so we can only eliminate the potential CMB risk in the model through statistical control.

**Table 10.** Descriptive Statistics of SEM.

| Variable | Define | Mean | Std. Dev. | Min | Max | |
|---|---|---|---|---|---|---|
| Local_feel | The feeling of being local | 1.224 | 0.417 | 1 | 2 | 1. Don't have. 2. Have |
| Part_feel | The feeling of being a part of the city | 3.26 | 0.649 | 1 | 4 | 1–4, The intensity of feeling increases with the numbers |
| Memb_feel | The feeling of being a number of the city | 3.215 | 0.679 | 1 | 4 | 1–4, The intensity of feeling increases with the numbers |
| INC | Gross monthly income level | 5.287 | 2.767 | 1 | 10 | 1–10, Income increases with the numbers |
| CONS | Total monthly consumption level | 5.269 | 2.902 | 1 | 10 | 1–10, Consumption increases with the numbers |
| Family size | Total household population | 2.869 | 1.226 | 1 | 9 | |
| Health | Health status | 3.764 | 0.974 | 1 | 5 | 1–5, Health improves as numbers increase |
| Education | Education | 3.484 | 0.988 | 1 | 7 | 1. No school education. 2. Elementary school. 3. middle school. 4. High school. 5. College degree in specialty. 6. General college degree. 7. Postgraduate. |
| Gender | | 1.448 | 0.497 | 1 | 2 | 1. Female. 2. Male |
| Age | | 32.87 | 8.72 | 15 | 60 | |
| Hukou | Household registration ageattributes | 0.139 | 0.346 | 0 | 1 | Agricultural Hukou = 0, Non-agricultural Hukou = 1. |
| Married | Marital status | 0.741 | 0.438 | 0 | 1 | Married = 1, Other situations = 0. |
| Medicinsur | Medical insurance program | 0.879 | 0.326 | 0 | 1 | Participate in any medical insurance program, yes = 1, No = 0. |
| Socialsecur | Social insurance program | 0.751 | 0.433 | 0 | 1 | Participate in any social insurance program, yes = 1, No = 0. |

Source: Developed by the author based on CMDS data.

Referring to the solution suggested by Podsakoff et al. (2003) [83], this paper first constructs Harman's single-factor test to determine the presence of CMB. Specifically, including all observed variables into an exploratory factorial analysis and examining the unrotated factor to know the number of factors with eigenvalues in excess of 1. For any factor with an eigenvalue greater than 1, this explains more than 50% of the covariance between the items and the standard components indicates the presence of CMB [82]. Exploratory factorial result presents in Table 11 shows that eigenvalues of Factor 1 and Factor 2 are greater than 1 and neither account for more than 50% [84]. In short, the Harman's single-factor test shows that there is no potential CMB problem.

In addition, this paper further examines the CMB by controlling for the effects of an unmeasured latent methods factor. Specifically, the latent variable CMB is measured with all observed variables, and then the latent variable CMB is fitted by substituting the original model, and finally, the fit of the original model is compared with the extended model to determine whether there is a CMB problem [83]. Table 12 shows the fitted situation of the two models, and it can be found that the original model fit is significantly better than the extended model. In summary, it can be concluded that there is no potential CMB problem in this study.

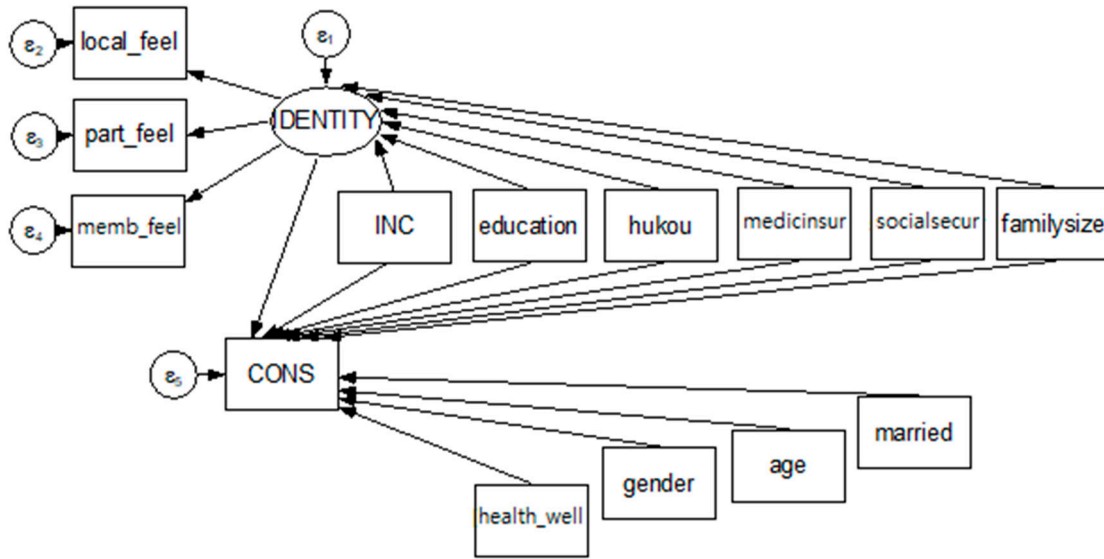

**Figure 5.** Structural equation modeling path framework. Source: Developed by the author based on CMDS data and STATA.

**Table 11.** Harman's single-factor test (CMB test).

| Factor | Eigenvalue | Difference | Proportion | Cumulative |
|---|---|---|---|---|
| Factor1 | 1.926 | 0.375 | 0.473 | 0.473 |
| Factor2 | 1.551 | 0.673 | 0.381 | 0.854 |
| Factor3 | 0.878 | 0.345 | 0.216 | 1.069 |
| Factor4 | 0.533 | 0.430 | 0.131 | 1.200 |
| Factor5 | 0.103 | 0.047 | 0.025 | 1.226 |
| Factor6 | 0.056 | 0.067 | 0.014 | 1.239 |
| Factor7 | −0.011 | 0.111 | −0.003 | 1.237 |
| Factor8 | −0.121 | 0.040 | −0.030 | 1.207 |
| Factor9 | −0.161 | 0.008 | −0.040 | 1.167 |
| Factor10 | −0.169 | 0.073 | −0.042 | 1.126 |
| Factor11 | −0.242 | 0.028 | −0.059 | 1.066 |
| Factor12 | −0.270 | | −0.066 | 1.000 |

Source: Developed by the author based on STATA.

**Table 12.** Unmeasured latent methods factor (CMB test).

| | RMSEA | CFI | TLI | SRMR |
|---|---|---|---|---|
| Basic model | 0.040 | 0.977 | 0.960 | 0.025 |
| CMB model | 0.114 | 0.812 | 0.720 | 0.083 |
| Threshold value | <0.05 | >0.9 | >0.9 | <0.05 |

Source: Developed by the author based on STATA.

### 5.4. Host Identity Measurement

After exploratory factor analysis, this paper selects three indicators with strong correlations, which are perceived to be local, perceived to be part of the city, and perceived to be a member of the city to measure the factor identity. The factor model was set as follows.

$$Y = \omega y IDENTITY + \xi \tag{5}$$

$Y$ is a vector of observed variables, where $y_1$ indicates that one considers oneself as a local, $y_2$ that one considers oneself as part of this city, and $y_3$ that one considers oneself as a member of this city. These three indicators together measure the identity of the migrant. The correlation between the explicit and latent variables is represented by the matrix $\omega_y$.

The results of the factor analysis and identity measures are given in Table 13. Part A of Table 13 shows that the Cronbach's Alpha is 0.7127 (>0.7), indicating three measured variables are internally consistent. Additionally, Kaiser–Meyer–Olkin (KMO) is 0.7 (>0.5), and Barlett's test of sphericity indicated a significant value ($p < 0.001$). Therefore, the use of factor analysis is appropriate for this study. Part B of Table 13 shows that only factor 1 has an eigenvalue greater than 1, indicating that the three measurement variables indeed measure only one unique common factor, host city identity. As can be seen from part C of Table 13, the root mean square error of approximation (RMSEA) is 0, the comparative fit index (CFI) is 0.996, the Tucker–Lewis index (TLI) is 0.991, and the SRMR is 0, which indicates a good fit [85].

**Table 13.** Host city identity measurement.

| Part A | Factor Analysis Test | | | |
|---|---|---|---|---|
| | **Bartlett Test** | **Bartlett (*p* Value)** | **KMO** | **Cronbach's Alpha** |
| | 1,092,749.96 | 0.000 | 0.700 | 0.7127 |
| **Part B** | **Factor Analysis** | | | |
| **Factor** | **Eigenvalue** | **Difference** | **Proportion** | **Cumulative** |
| *Factor1* | 1.473 | 1.490 | 1.145 | 1.145 |
| *Factor2* | −0.017 | 0.152 | −0.013 | 1.132 |
| *Factor3* | −0.169 | | −0.132 | 1.000 |
| **Part C** | **Host City Identity Measurement** | | | |
| **Measurement (Standardized)** | **Coefficient** | **std. err.(OIM)** | **z** | ***p* > |z|** |
| *IDENTITY* | | | | |
| *local_feel* | 0.276 *** | 0.008 | 33.90 | 0.000 |
| *part_feel* | 0.869 *** | 0.010 | 90.28 | 0.000 |
| *memb_feel* | 0.896 *** | 0.010 | 91.08 | 0.000 |
| RMSEA = 0.000 CFI = 0.996 TLI = 0.991 SRMR = 0.000 CD = 0.879 ρ = 0.846 | | | | |

Note: *** $p < 1\%$.

## 5.5. Empirical Results

This work uses the Maximum Likelihood Estimation (MLE) to estimate the effect of host city identity on migrants' consumption. The estimation results are shown in Table 14. The values of CFI and TLI are greater than the critical values (0.9). The values of RMSEA and SRMR are less than the critical values (0.05). It indicates that the model is acceptable. Specifically, host city identity has a significant positive effect on consumption, and for each unit increase in identity, the consumption will rise by 5.2%. Education, income, Hukou, and social insurance all have a significant positive effect on consumption. The marginal effects of education and income on consumption are larger than the effects of hosting urban status. This suggests that to improve consumption inequality among migrants it is necessary to increase their sense of urban belonging and promote their income. The effects of Hukou and social insurance programs on consumption are in line with theoretical expectations that having an urban Hukou or participating in social insurance programs will increase consumption. It is worth noting that education, Hukou, and social insurance programs are important influences in the process of host city identity construction.

**Table 14.** Structural equation modeling results.

| | **Standardized Coefficient** | **Unstandardized Coefficient** | **z** | **p > |z|** |
|---|---|---|---|---|
| **Structural** | | | | |
| *CONS* | | | | |
| *IDENTITY* | 0.052 *** | 1 *** | 8.710 | 0.000 |

**Table 14.** *Cont.*

|  | Standardized Coefficient | Unstandardized Coefficient | z | p > |z| |
|---|---|---|---|---|
| *education* | 0.084 *** | 0.247 *** | 12.880 | 0.000 |
| *hukou* | 0.048 *** | 0.401 *** | 7.850 | 0.000 |
| *medicinsur* | −0.019 *** | −0.165 *** | −2.820 | 0.005 |
| *socialsecur* | 0.021 *** | 0.143 *** | 3.200 | 0.001 |
| *INC* | 0.657 *** | 0.689 *** | 126.600 | 0.000 |
| *health_well* | −0.029 *** | −0.087 *** | −5.180 | 0.000 |
| *familysize* | 0.106 *** | 0.251 *** | 11.890 | 0.000 |
| *gender* | 0.012 ** | 0.070 ** | 2.170 | 0.030 |
| *age* | −0.016 ** | −0.005 ** | −2.260 | 0.024 |
| *married* | 0.027 *** | 0.182 ** | 3.010 | 0.003 |
| *_cons* | 0.080 | 0.234 | 1.590 | 0.111 |
| *IDENTITY* |  |  |  |  |
| *INC* | 0.016 | 0.009 | 1.590 | 0.111 |
| *education* | 0.056 *** | 0.019 *** | 5.560 | 0.000 |
| *hukou* | 0.043 *** | −0.003 *** | 4.560 | 0.000 |
| *medicinsur* | −0.006 | 0.023 | −0.590 | 0.555 |
| *socialsecur* | 0.067 *** | 0.001 *** | 6.410 | 0.000 |
| *familysize* | 0.014 | 0.002 | 1.320 | 0.186 |
| RMSEA = 0.040 SRMR = 0.025 CFI = 0.977 TLI = 0.960 SRMR = 0.025 | | | | |

Note: ** $p < 5\%$, *** $p < 1\%$.

The research on standardized path coefficients (Table 15 and Figure 6) shows that there is no significant effect of income on the construction of host city identity. In contrast, education, Hukou, and social security program all have a significant positive effect on the host city's identity. Specifically, the positive direct effect of social security on identity is greater, the direct effect of education has the second largest effect, and the direct effect of Hukou is the smallest. This suggests that, with the household registration system not yet abolished, increased investment in urban public services will help increase the migrants' sense of belonging. Meanwhile, the migrant needs to improve their education and acquire vocational skills to better integrate into the city. The total effect results indicate that education is the most important factor affecting the consumption of migrants other than income. In addition, the total effects of Hukou and host identity on consumption are similar, about 5%, indicating that both legal and psychological perceptions of host city identity can influence consumption behavior. For policymakers, removing the strict Hukou system and adopting open and inclusive management measures is a win–win way that not only promotes the integration of migrants into cities, but also stimulates urban consumption.

**Table 15.** Standardized path coefficient.

| Path | Direct Effects | Indirect Effects | Total Effects |
|---|---|---|---|
| *IDENTITY-> CONS* | 0.052 *** (0.000) |  | 0.052 *** (0.000) |
| *Education-> CONS* | 0.084 *** (0.000) | 0.003 *** (0.000) | 0.087 *** (0.000) |
| *Education-> IDENTITY* | 0.056 *** (0.000) |  | 0.056 *** (0.000) |
| *INC-> CONS* | 0.657 *** (0.000) | 0.001 (0.117) | 0.658 *** (0.000) |
| *INC-> IDENTITY* | 0.016 (0.117) |  | 0.016 (0.117) |
| *hukou-> CONS* | 0.048 *** (0.000) | 0.002 *** (0.000) | 0.050 *** (0.000) |

**Table 15.** *Cont.*

| Path | Direct Effects | Indirect Effects | Total Effects |
|---|---|---|---|
| *hukou-> IDENTITY* | 0.043 *** (0.000) | | 0.043 *** (0.000) |
| *socialsecur-> CONS* | 0.021 *** (0.001) | 0.003 *** (0.000) | 0.024 *** (0.000) |
| *socialsecur-> IDENTITY* | 0.067 *** (0.000) | | 0.067 *** (0.000) |

Note: *p*-value in the parentheses. *** *p* < 1%.

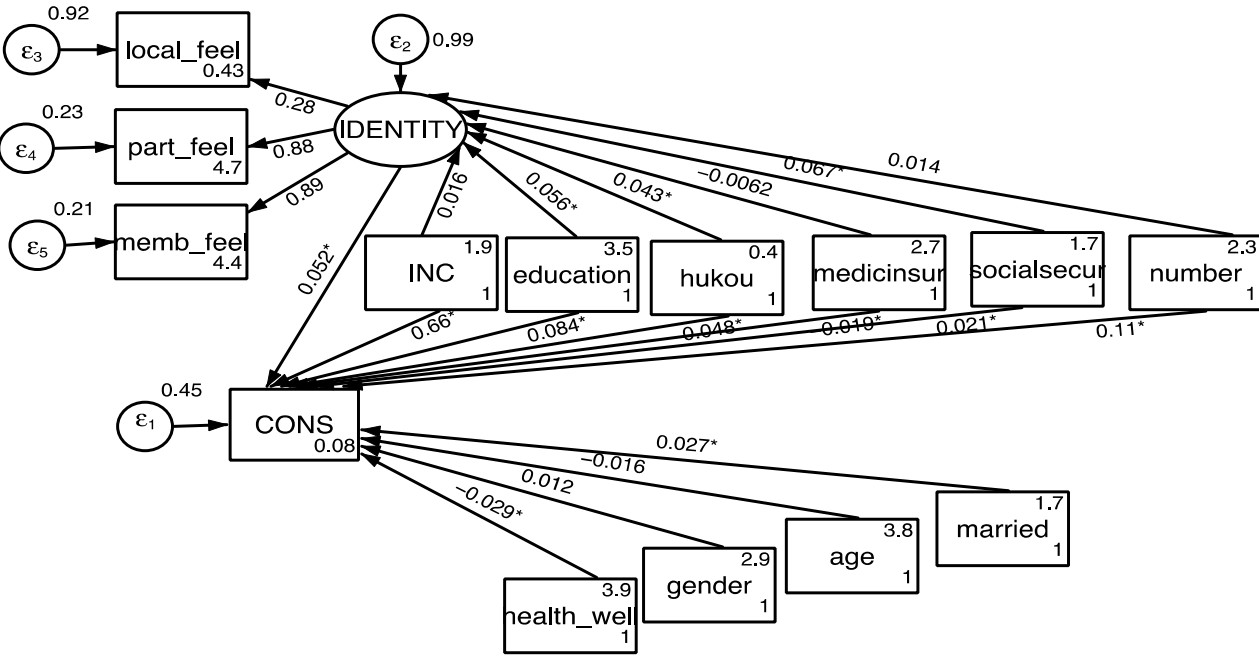

**Figure 6.** Standardized path coefficient. Source: Developed by the author based on CMDS data and STATA. * Indicates significant at the 0.1% significance level.

## 6. Conclusions

Following this research, we concluded that host city identity has a significant impact on the consumption behaviors of rural–urban migrants. As indicated by the OLS empirical results, compared with those without a host city identity, the household consumption and savings of migrants with a host identity increased by 4% and decreased by 1.7%, respectively. The SEM results show that a one-unit increase in host identity will increase the consumption by 5.2%. It should be noticed that income and education have a greater impact on consumption than host identity. In addition, education, household registration, and social insurance have a greater impact on host identity. In conclusion, this study shows that the consumption of the rural–urban migrant populations has a significant host city identity effect. Education, household registration, and social insurance have a significant impact not only on consumption but also on host identity. In conclusion, this study shows that the consumption of the rural–urban migrant population has a significant effect on the identity of the host city. Education, household registration, and social insurance affect consumption levels as well as the host identity perceptions of the migrant.

### 6.1. Theoretical Implications

This work aims to add to the literature on host city identity effects in the field of the consumption of rural–urban migrants in China. The empirical method used in this paper facilitates both the analysis of the consumption of the rural–urban migrants and the understanding of the process of constructing of their host identity.

This paper also enriches the literature on the psychological and economic integration and consumption inequalities of rural–urban migrants in China. To break the inequality of consumption of migrants, the focus should be on the perception of identity and their employment income. Equal employment opportunities are necessary for both the income and consumption of the rural–urban migrants. Removing household registration controls and increasing the supply of public services to the migrants can increase both the sense of belonging and the consumption.

### 6.2. Managerial Implications

Over the past few decades, the number of rural–urban migrants in China has been rising rapidly. At the end of 2017, the total number reached 241 million, having exceeded 15% of the total population of China. Rural–urban migrants have made huge contributions to the urban economy. Moreover, they have brought enormous challenges to the city managers (e.g., their welfare and integration in the urban).

The empirical findings here are of significant imply policy significance to the managers of China's cities. The inequality currently encountered by migrants in cities has been widespread (e.g., inequality of income, inequality of stable job opportunities, inequality of consumption, and inequality of social benefits and public services). City managers should reduce rural–urban segregation by opening more public service accesses to migrants, offering more suitable and affordable housing, and reforming the rigorous Hukou registration system. The above measures help migrants form a sense of belonging, promote their integration into urban society, and ensure urban harmony and stability. In addition, it increases the consumption and welfare of immigrants and also solves the challenge of low consumption ratio and promotes the economic development of the city.

### 6.3. Limitations and Further Research

This study has some limitations. First, the scope of this paper is limited to China, therefore, the results are not generalizable. Second, identity perceptions, income, and consumption were self-reported by respondents, and there is a risk of misreporting data by observers. Third, the pathway constructed in this paper to simplify the analysis did not consider the effect of demographic characteristics on identity. Fourth, this paper uses secondary survey data, which does not allow for quality control of the data generation process. In further research, using objective data rather than self-reported data to measure identity, income, and consumption may increase the accuracy of conclusions. Data quality and collection procedures can be controlled through field investigation as compared to the direct use of secondary survey data.

In order to have an all-round understanding of the psychological integration and economic integration of the migrant population in China, this study tries to explain the influence of host city identity on consumption behavior of the migrant population and also to explore the factors influencing host city identity. The findings of this work not only help to solve the problem of low consumption among the migrant population in China but also help to promote the economic and psychological integration of the migrant population in the city. In the process of urban integration of the migrant population, policymakers and city managers should not only liberalize household registration, alleviate employment inequality and increase the supply of public services, but also implement more humanistic care for the migrant population so that they can complete their identity transformation.

**Author Contributions:** Conceptualization, N.M. and W.S.; methodology, W.S.; software, N.M. and Z.W.; validation, N.M. and W.S.; formal analysis, N.M. and Z.W.; investigation, N.M.; resources, W.S.; data curation, N.M.; writing—original draft preparation, N.M.; writing—review and editing, W.S.; visualization, N.M.; supervision, W.S.; project administration, W.S.; funding acquisition, W.S. All authors have read and agreed to the published version of the manuscript.

**Funding:** This research was funded by the National Natural Science Foundation of China (NO. 71903210) and Beijing Social Science Fund Project (NO. 21DTR010).

**Institutional Review Board Statement:** Not applicable.

**Informed Consent Statement:** Not applicable.

**Data Availability Statement:** Publicly available datasets were analyzed in this study. This data can be found here: https://www.chinaldrk.org.cn/wjw/#/application/userData (accessed on 1 August 2020).

**Conflicts of Interest:** The authors declare no conflict of interest.

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
