# Peer review of "Host Identity and Consumption Behavior: Evidence from Rural–Urban Migrants in China"

_sustainability, doi:10.3390/su141912462_

Round 1
Reviewer 1 Report
The article is interesting and the researched problem has scientific potential. However, some problems need to be solved:
1. The literature review should include more recent sources (2019-2022) and be enriched with relevant references.
2. The use of self-administered questionnaires can generate a problem that may affect the relevance of the research: the bias effect or common method bias - CMB (see: Podsakoff PM, MacKenzie SB, Lee, JY, Podsakoff NP. Common method biases in behavioral research: A critical review of the literature and recommended remedies. Journal of Applied Psychology. 2003; 88(5):879-903.). Such problems arise when data on independent and dependent variables emanate from the same respondent, and the same measurement scale exists throughout the questionnaire. Authors must take action to prevent common method bias - CMB (e.g., https://doi.org/10.3390/ijerph182312387 or https://doi.org/10.3390/systems10040121)
3. The article would gain value if the authors will use SEM to highlight the relationships among variables.
4. There is a need for a discussion section in which authors must be built in the context of dialogue with other researchers in the literature review.
5. In my opinion, a section of conclusions that includes theoretical and managerial implications, research limitations, and future research directions would be helpful.
The article has scientific value and can be published after carefully reviewing the reported issues.
Reviewer 2 Report
This paper examines the impact of host identity on household consumption for migrants in China. Host identity is measured by a Y/N question of whether he/she is local. It finds that host identity significantly impacts rural-urban migrants' consumption. The paper is interesting, has a good story and multiple empirical testing, but I have some suggestions that can possibly improve the paper's quality:
1. The structure of the abstract is confusing. It would be better if it could simply be: purpose - data - finding - implication. Indeed, the authors can add an explanation of the host identity in one or two sentences.
2. The paper investigates whether consumption is impacted by host identity. Therefore, I am not sure that the variable "Saving rate" is relevant here as the dependent variable. One dependent variable, which is the log of consumption, is enough. The authors could move the saving rate into the robustness checks.
3. How is it possible that Rev identity has a different sign than other identity variables? What is the explanation?
4. The instrumental variable regression (2SLS) is only valid if the authors present in Table: (1) the F-test of the first stage regression and (2) the Hansen test. The first test has been provided by the authors, but the second test is missing. Without Hansen test, we do not know whether the instruments are valid.
6. The authors should provide robustness checks.
7. The English should be improved. Copyediting by a native might be needed.
Reviewer 3 Report
Dear Authors!
Thanks for the interesting article.
1) It is appropriate to finalize the abstract. The abstract must include sufficient information for readers to judge the nature and significance of the topic. The abstract should contain the main idea of the paper, the subject and the goal of the research, methods used, hypotheses, research results and a brief conclusion.
2) Keywords should be refined. Keywords should not exclusively duplicate the title of the article.
3) I recommend that in the conclusions, you mention the results of mathematical calculations, which are given in the article.
Best regards
Round 2
Reviewer 1 Report
The paper can be published in the current version.